# Positive Effects of Land Use Change on Wintering Bar-Headed Geese between 2010 and 2021

**DOI:** 10.3390/ani12223142

**Published:** 2022-11-14

**Authors:** Zhongrong Wu, Xiulin Ye, Zhongfan Kuang, Hui Ye, Xumao Zhao

**Affiliations:** 1School of Agriculture, Yunnan University, Kunming 650504, China; 2Wildlife Protection Station, Guiyang Forestry Bureau, Guiyang 550007, China; 3College of Ecology, Lanzhou University, Lanzhou 730000, China; 4Guizhou Institute of Biology, Guiyang 550009, China

**Keywords:** waterbirds, population size, *Anser indicus*, satellite tracking, habitat selection

## Abstract

**Simple Summary:**

Changes in land use caused by human activities can potentially result in population decline and/or local extinction. However, some species can derive benefits from particular land use changes. To understand the effects of land use change on individual species, we investigated the wintering population of bar-headed geese for the past 11 years in Caohai Guizhou Province in China. Wintering bar-headed geese were fitted with satellite trackers to assess their use of different land types and the impact of land use changes occurring between 2010 and 2021. Our study reveals that the wintering population size of bar-headed geese increased from 1366 to 2864 and recent land use changes have had a positive impact on waterbird populations. Our study provides a case study for managing human–wildlife relationships and protecting waterbirds and other wildlife.

**Abstract:**

Human-induced land use change often drives species losses, yet some species can derive benefits from particular land use changes. Thus, case studies of how specific land use changes affect population size for species of interest are essential to their conservation. In this study, wintering bar-headed geese in Caohai, in Guizhou Province in China, were fitted with satellite trackers to assess their use of different land types and the impact of land use changes occurring between 2010 and 2021. We found that bar-headed geese preferentially spent time in arable lands, grasslands, and open water; most foraging occurred in cropland (59.5%) and grasslands (26.4%), while resting occurred in open water (68.3%) and in grasslands (43.5%). The population of wintering bar-headed geese in Caohai increased in size from 1366 to 2803 between 2010 and 2021. A concomitant decrease in cropland area (10.7%) and increase in open water (5.52%) and grasslands (48.45%) positively affected population growth. The use of abandoned croplands reduced human disturbance of goose foraging, while larger water and grassland areas provided more foraging and resting opportunities for bar-headed geese. Our study reveals a positive impact of recent land use changes on waterbird populations and provides a case study for managing human–wildlife relationships and protecting waterbirds and other wildlife.

## 1. Introduction

Land use change is an important driver of global biodiversity loss, and human-dominated habitats also have lower species diversity and more homogeneous communities [1,2]. Land use changes caused by human activities, such as the construction of roads, settlements, and urban areas, lead to habitat fragmentation for wildlife, potentially resulting in range reductions and/or local extinction [3]. However, not all species are adversely affected by human-mediated land use change; for example, some species may expand their ranges into the novel habitats produced [4,5]. Shifting land use patterns can alter community structure and interspecific relationships, as well as create novel habitats that may nurture particular species [6,7]. Therefore, understanding the effects of land use change on individual species is critical, not only for conservation efforts, but also for assessing ecosystem-level effects of land use change [5].

To meet demands for human food, China has long converted forests, grazing lands, and wetlands into farmland, resulting in severe ecological degradation [8,9]. Between 1950 and 2000, for example, 65% of the wetlands associated with the Bohai Sea and Yellow Sea disappeared [10], negatively impacting waterbird populations and diversity. To mitigate the ecological effects of these historical land use patterns, the Chinese government implemented a series of ecological restoration projects in the past 20 years, including afforestation, the return of farmland to lakes, and the establishment of protected areas [11]. However, changes in land use can affect individual species differently, creating both “winners” and “losers”. For example, the aquaculture industry provides food and alternative habitats for 69% of China’s waterbirds; however, threatened species benefit less from aquaculture than unthreatened species [12]. Therefore, studying the impacts of land use changes on individual waterbird species is essential for their conservation.

China’s wetlands support the breeding and wintering of millions of waterbirds. Caohai, a wetland on the Yunnan–Guizhou Plateau, is an important wintering ground for waterbirds such as bar-headed geese (*Anser indicus*). Bar-headed geese breed in the high-elevation wetlands of Central Asia, which range from central China in the east to Tajikistan in the west, to southern Russia in the north, and to India in the south [13]. In China, bar-headed geese breed in the lakes and marshes of the Qinghai and Tibetan plateaus, then overwinter in Yunnan, Guizhou, and southern Tibet [14]. In some studies, agricultural activities have been found to negatively affect bar-headed goose populations, as geese are chased from crop fields by farmers [15]. However, other studies have found that the increase in farmland in Tibet, and especially the planting of winter wheat, has supported the growth of the wintering population of bar-headed geese [13]. Here, we study the effects of land use change on waterbird populations, taking bar-headed geese as an example. We used direct counts, satellite tracking, and behavior observation to explore how bar-headed geese utilize different land types and the impact of land use changes on their population dynamics.

## 2. Materials and Methods

### 2.1. Study Area and Investigation

This study took place in Caohai (N 26°47′32″–26°52′52″, E 104°10′16″–104°20′40) in Weining County, Guizhou Province, China. Caihai has an altitude of 2200–2500 m and covers an area of 96 km^2^, including 46.5 km^2^ of water, with an average water depth of 1.5 m [16]. The climate is subtropical monsoon, with an average annual temperature of 10.5 °C and an average annual precipitation of 950.9 mm [15]. In addition to bar-headed geese (*Anser indicus*), waterbirds that regularly overwinter in Caohai include spot-billed ducks (*Anas poecilorhyncha*), ruddy shelducks (*Tadorna ferruginea*), black-necked cranes (*Grus nigricollis*), and common cranes (*Grus grus*).

We systematically surveyed wintering populations of bar-headed geese at seven sites in Caohai every month during the wintering season for 11 consecutive years (2011–2021). The surveys were facilitated by the fact that bar-headed geese congregate each night at a roost site. Bar-headed geese overwinter in Caohai from November to March each year. They have large population changes in November and March, and are more stable in other months. The seven roost sites we identified in Caohai were: Huyelin, Liujiaxiang, Suohuangchang, Wangjiayuan, Wenjiatun, Wujiayan, and Yangguanshan (Figure 1). We counted bar-headed geese at night using monoculars (20–60) and binoculars (10 × 42) in December–January of every year between 2011 and 2021. Our investigation team consisted of seven survey groups of 2–3 people.

### 2.2. Habitat Selection and Time Allocation

We used two kinds of satellite trackers (Anit-GT, 0325 25 mm × 60 mm × 29 mm, weight 25g; HQBG2512S, 24 mm × 48 mm × 28 mm, weight 14 g) to track the bar-headed geese. Following national standards, satellite trackers must account for less than 3% of the target animal’s body weight. The satellite trackers selected represent 1% and 0.56% of a bar-headed goose’s body weight (about 2.5 kg), respectively, which should not affect normal activities. GPS signal points were received every 1–3 h. There are five levels of accuracy for satellite trackers: A (<5m), B (5 m < 10 m), C (10 m < 20 m), D (20 m < 100 m), and E (100 m < 2000 m). To ensure accurate results in our surveys, only levels A and B were used in our analysis. We put satellite trackers on 19 bar-headed geese from December 2017 to January 2019, attaching the trackers to the backs of the geese. In total, we obtained 13,013 data points; the habitat utilization of each bar-headed goose was judged according to their tracked locations. We also calculated the intensity of habitat selection by bar-headed geese, where intensity was defined as the frequency divided by the habitat area.

### 2.3. Behavior Observation

Food is a reuse factor in wintering habitat selection for waterbirds. We observed and quantified the feeding behavior of bar-headed geese. We used telescopes to observe behavior in the daytime during the wintering season. We selected bar-headed geese flocks with no less than 30 individuals to observe, scanned these flocks and recorded behaviors every five minutes, randomly observed spontaneous behaviors, and recorded the habitat type. We used the following definitions of bar-headed geese behaviors:Foraging: head down for food, turning head to look for food, walking slowly and eating with head down, and thrusting the head into the water to find food, then raising the head and swallowing;Resting: one or two legs in grass or water with the neck constricted into an “S” shape against the body, or head turned back with the beak inserted under the wing;Vigilance: standing still and looking around with head up;Preening: combing the feathers, tarsi, and feet with the front end of the beak, and/or rubbing grease;Locomotion: walking, flying, jumping, and flapping;Social behaviors: threatening, attacking, avoiding, chirping, and fighting.

### 2.4. Data Analyses

To quantify land use changes in Caohai between 2010 and 2021, we obtained historical land use data for Caohai from remote sensing data (https://www.tianditu.gov.cn/ (accessed on 7 September 2022). In ArcGIS 10.5, three land use types were delineated in Caohai: croplands, grasslands, and open water. We also extracted temperature, precipitation, and wind velocity data for 2001 to 2021 from China’s National Meteorological Administration (http://www.cma.gov.cn/, accessed on 7 September 2022).

We used multiple linear regression and univariate regression to assess how climatic factors and land use changes affected bar-headed goose populations. We quantified the impact of the croplands, grasslands, open water, temperature, precipitation, and wind velocity data on bar-headed geese population trends over an 11-year period (2001–2021) using generalized models (Appendix A). To avoid model overfitting, we used Spearman’s rank correlation coefficients the relationships among these six factors and variance inflation factor (VIF) (Appendix A and Appendix A). We only retained one factor when the Spearman’s rank correlation coefficient was >0.8 [17,18]. Then, we used model selection to find the factor affecting the population’s size trend. We found the best model among subset models based on small sample sizes (AICc) using the dredge function in the “MuMIn” package [19]. We also analyzed the effect of each factor on population trends of bar-headed geese separately.

## 3. Results

### 3.1. Change in Population Size of Bar-Headed Goose Population

We found that the bar-headed goose population in Caohai experienced significant growth from 2011 to 2021. Bar-headed geese numbered 2864 in 2021, more than twice the number in 2011 (1366) (Table 1). Huyelin, Yangguanshan, and Wenjiatun were the main roost sites for bar-headed geese, accounting for 60.9% of the population in Caohai (Table 1).

### 3.2. Habitat Selection by Bar-Headed Geese

We recorded a total of 45,920 bar-headed geese through satellite tracking, of which 73.4% occurred in croplands, followed by grasslands (12.7%), and open water (13.9%) (Figure 1). Bar-headed geese were not found in buildings, woodlands, or on roads. The highest habitat selection intensity was recorded for bar-headed geese in grasslands (2.3), followed by croplands (1.3), and open water (1.2).

### 3.3. Time Allocation for Behaviors

When in cultivated lands, bar-headed geese spent 59.5% of their time on feeding behavior and 34.3% on resting behavior, while in grasslands, 43.5% of their time was spent on feeding behavior and 26.4% on resting behavior (Figure 2). Finally, on the open water, 68.3% of time was spent on resting behavior, 13.4% on locomotion, and 2.4% on feeding behavior.

### 3.4. Change in Land Use

After model selection, the linear models showed that the increase in the open water area had a positive effect on the population increase of wintering bar-headed geese over the past 11 years (Figure 3; Appendix A). When the effect of each factor was analyzed separately, we found the size of grassland and open water areas was positively correlated with the bar-headed goose population size (*p*< 0.05), while the cultivated land area was negatively correlated with population size (*p* < 0.05) (Figure 4). Both single factor analysis and linear analysis showed that climatic factors had no significant effect on the population size of wintering bar-headed geese. We also found grasslands expanded by 48.45% in Caohai; the area covered by water also increased by 5.52%, while the cropland area decreased by 10.7% between 2010 and 2021 (Appendix A).

## 4. Discussion

The size of the wintering population of bar-headed geese increased by 105% between 2010 and 2021 in Caohai (Table 1). This finding is consistent with the more general trend observed for bar-headed geese in south-central Tibet, where the wintering population has increased more than four times in size between 1993 and 2014 [13]. However, Birdlife International [20] estimates that bar-headed goose populations are declining on a global scale. Therefore, the local increase in the Tibetan population may be due to the improvement of wintering habitat [13].

Using satellite tracking, we accurately assessed habitat use in wintering bar-headed geese in Caohai, improving on more traditional observation methods. We found that the main habitat types used by the bar-headed geese were croplands, grasslands, and open water, of which croplands were the most important foraging ground. In south-central Tibet, wintering bar-headed geese mostly fed within winter wheat fields [13], consuming plant leaves and stems, as well as the seeds of leguminous plants [14]. In Caohai, wintering bar-headed geese mainly fed on crops, especially those found in vegetable fields in January, and 65% of their food consisted of Gramineae species leaves [21].

Human activity can interfere with foraging, but arable land provides many bird species with food [22]. We found that the decrease in cropland area between 2010 and 2021 had a positive effect on the bar-headed goose population. This differs from findings for south-central Tibet, where an increase in the area planted with winter wheat positively affected the bar-headed goose population [13]. In Tibet, the local people believe in Buddhism and protect animals from harm, especially bar-headed geese, as they are mascots of Buddhism [13]. Outside of Tibet, local residents may chase, intimidate, and even poison foraging birds to protect their farms. With the emergence of new farming methods, competition for land between humans and birds has become increasingly severe, seriously affecting the survival of wintering waterbirds [23]. Here, the decline in cropland area was due to the abandonment of former croplands. Abandoned farmlands provide space for grasses and sedges to grow, which then serve as a rich food source for bar-headed geese. In Caohai, the local government has systematically mandated local residents to abandon croplands since 2013. Thus, the increase in croplands in Caohai provided food resources for wintering cranes.

We also found that rising water levels had a positive effect on wintering bar-headed geese (Figure 3 and Figure 4). Caohai is an important wintering ground for bar-headed geese on the Yunnan–Guizhou Plateau. In Caohai, the Chinese government has invested CNY 10.9 billion in environmental remediation since 2015, including water replenishment initiatives. Given that bar-headed geese forage around water, the expansion of wetlands (caused by rising water levels) has likely provided more foraging grounds for bar-headed geese [24].

In addition, the bar-headed geese wintering in Caohai mainly breed in Zoige. Between 2000 and 2019, the Zoige wetlands were generally improved via several ecological restoration projects and protection management [25], providing more stable conditions for bar-headed goose breeding. Caohai National Nature Reserve has strengthened wetland management since the promulgates of the ‘Regulations on Wetland Protection in Yunnan Province’ in 2013. These factors will also play a positive role in increasing the wintering population of bar-headed geese.

## 5. Conclusions

Our study reveals how land use changes have affected wintering bar-headed goose populations. A decrease in cropland area and simultaneous increase in grassland and open water areas promoted the growth of the wintering bar-headed goose population. However, we found that climate had little effect on the goose population between 2001 and 2021. Climate change is a global change driver, leading to shifts in species distributions and population sizes. However, the survey period used here was relatively short (and the geographic scope small), potentially limiting our ability to detect any climate change effects. In addition, any changes to the bar-headed goose population in the breeding site may also affect the wintering population, but we lacked survey data for the breeding site.

## Figures and Tables

**Figure 1 animals-12-03142-f001:**
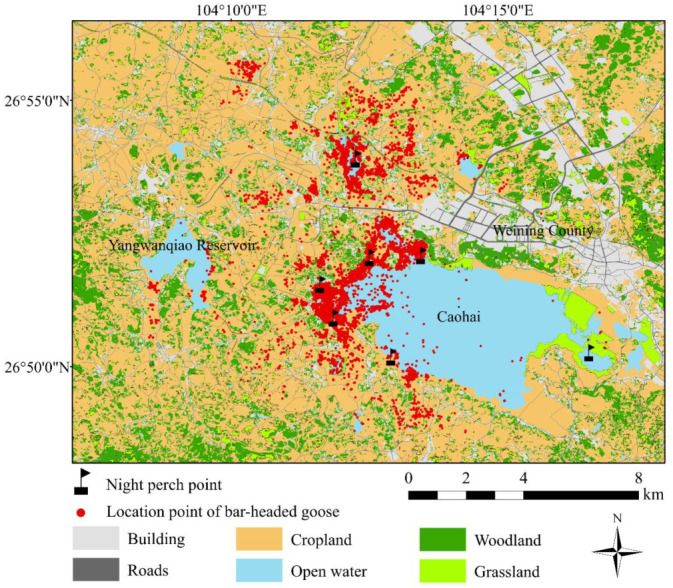
Location of 19 individual bar-headed geese derived from satellite tracking over a 13-month period in Caohai, with land use types indicated on the map.

**Figure 2 animals-12-03142-f002:**
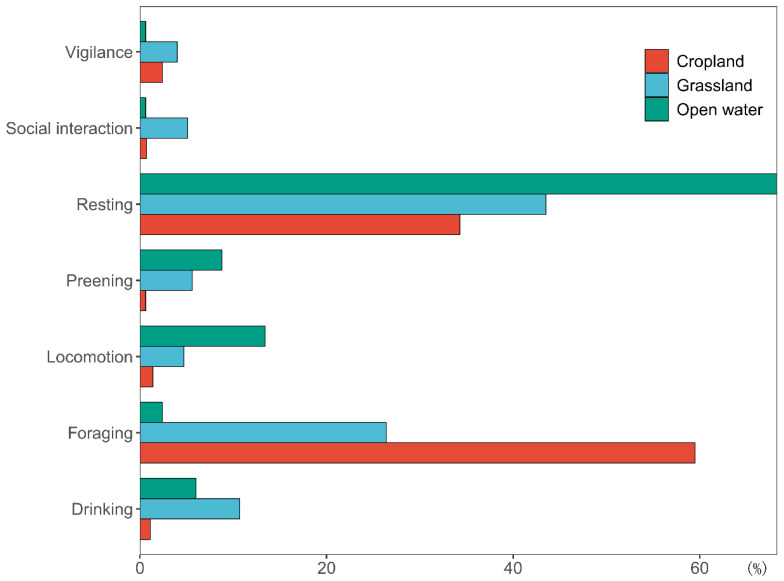
Daytime behaviors of wintering bar-headed geese in different habitat types.

**Figure 3 animals-12-03142-f003:**
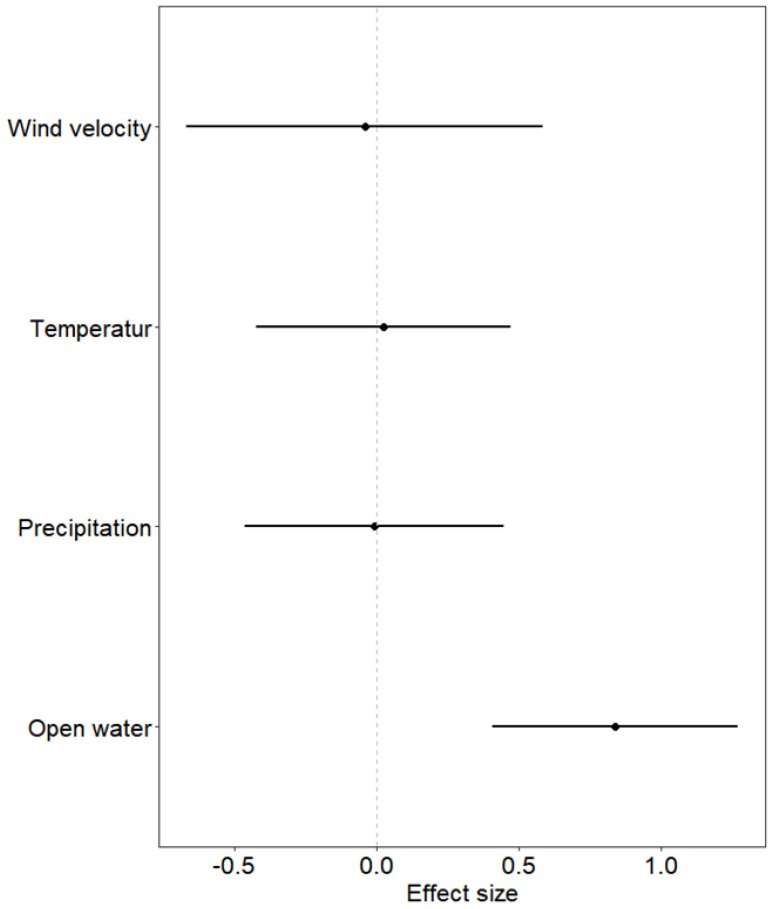
Effect sizes of Precipitation, Temperature, Open water, and Wind velocity on population trend of bar-headed geese. Effect sizes are weighted average standardized coefficients across models with ∆AICc < 6. The lines represent 95% confidence interval (CI).

**Figure 4 animals-12-03142-f004:**
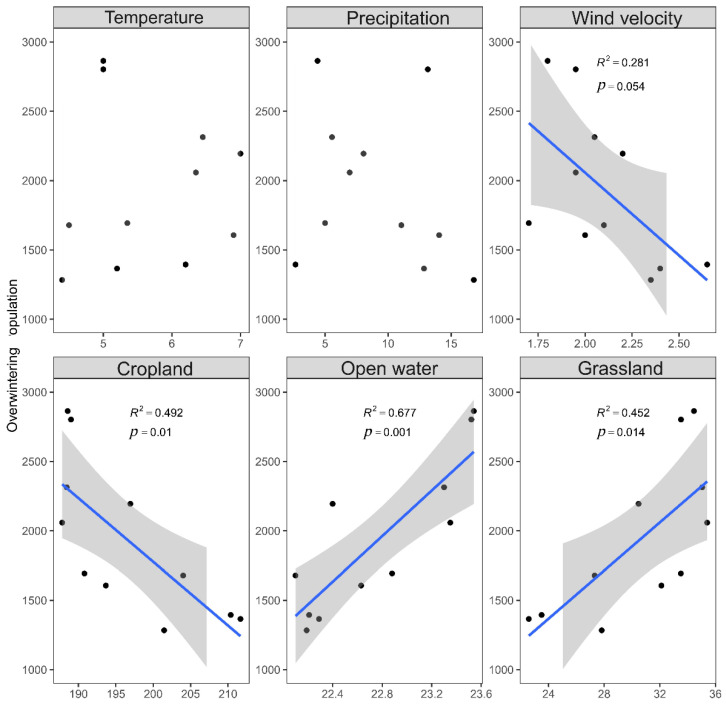
The effect of Cropland, Grassland, Precipitation, Temperature, Open water, and Wind velocity on wintering population size of bar-headed geese over 11 years.

**Table 1 animals-12-03142-t001:** Bar-headed goose population size in Caohai from 2011 to 2020.

Year	Site 1	Site 2	Site 3	Site 4	Site 5	Site 6	Site 7	Total
2011	482	124	220	96	156	148	140	1366
2012	651	365	125	102	85	0	67	1395
2013	339	406	221	84	239	216	174	1679
2014	425	128	350	72	189	120	0	1284
2015	492	395	356	258	301	209	184	2195
2016	387	201	227	114	423	113	142	1607
2017	264	198	354	93	228	370	187	1694
2018	529	325	598	284	329	102	147	2314
2019	412	293	612	155	109	392	86	2059
2020	524	473	763	462	266	376	0	2864
2021	984	118	284	112	409	823	73	2803

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
