# Peer review of "Positive Effects of Land Use Change on Wintering Bar-Headed Geese between 2010 and 2021"

_animals, 2022, doi:10.3390/ani12223142_

Round 1
Reviewer 1 Report
This manuscript showed that the human-induced changes of land use caused a large increase of bar-headed goose population. The study is based on field surveys, satellite tracking, and habitat classification using remote sensing images. The data are reliable, and the results are meaningful, representing the latest status of the bar-headed goose and human activities in an important nature reserve, Caohai. Given the negative effects of land use changes to most wildlife, this study indicated the complexity of human-wildlife relationship. I have two major comments listed below:
1. The main objective of this study is to demonstrate the effects of land use change on bar-headed goose population. The bar-headed goose population (temporal dynamics and spatial distribution) is well introduced. However, the changes of land use were missing. Please add this information, such as the dynamics of croplands areas from 2010 to 2021.
2. Six variables (croplands, grasslands, open water, temperature, precipitation and wind velocity) were used to explain the dynamics of population sizes from 2010 to 2021. The nature of the six variables should be provided in a table, including this mean values, minimum and maximum values, units, etc.
I also have some minor comments:
1. Line 25. “most foraging occurred in fields (59.5%)”. What is “fields”. You may mean croplands.
2. Line 90. After listing the seven roost sites, you’d better insert the word “(Figure 1)” here, so that readers can find the sites in the figure.
3. Lines 125-126. “six land use types were delineated in Caohai: croplands, grasslands and open water.”. I only saw three types.
4. Lines 132-133. “To avoid model overfitting, we used spearman's rank correlation coefficients…”. Using correlation coefficients is not a good way for checking the independence of variables, because they only represent the relationship of two variables. The best method is using variance inflation factor (VIF), which quantifies the association of a variable with all other variables. The vif() function in R package “car” can do this job.
5. Line 153. The duration (starting day and ending day) of the tracked points should be provided. The number of tracked individuals should be introduced.
6. Figure 3 shows the effect size of the regression coefficient of open water areas. The areas of croplands maybe removed because of its high correlation with areas of open water. The authors should clarify this point. At line 172, change “Figure 4” to “Figure 3”.
7. Figure 4. The authors should clarify these are univariate analysis using 11 years as replications.
Author Response
Reviewer-1
This manuscript showed that the human-induced changes of land use caused a large increase of bar-headed goose population. The study is based on field surveys, satellite tracking, and habitat classification using remote sensing images. The data are reliable, and the results are meaningful, representing the latest status of the bar-headed goose and human activities in an important nature reserve, Caohai. Given the negative effects of land use changes to most wildlife, this study indicated the complexity of human-wildlife relationship. I have two major comments listed below:
Response: Thank you for these positive remarks. We have made every effort to respond to your concerns.
- The main objective of this study is to demonstrate the effects of land use change on bar-headed goose population. The bar-headed goose population (temporal dynamics and spatial distribution) is well introduced. However, the changes of land use were missing. Please add this information, such as the dynamics of croplands areas from 2010 to 2021.
Response: we modified the text on page 6 lines 178-179 as follow: “We also found grasslands expanded by 48.45% in Caohai; the area covered by water also increased by 5.52%, while the cropland area decreased by 10.7% between 2010 and 2021(Supplementary S1).”
Supplementary S1 the six variables including the croplands, grasslands, open water, temperature, precipitation and wind velocity.
Year |
Temperature (℃) |
Precipitation (mm) |
Wind velocity (m/s) |
Cropland (km2) |
Open water (km2) |
Grassland (km2) |
2010 |
5.2 |
12.85 |
2.4 |
211.67 |
22.29 |
22.58 |
2012 |
6.2 |
2.65 |
2.65 |
210.37 |
22.21 |
23.5 |
2013 |
4.5 |
11.05 |
2.1 |
204 |
22.1 |
27.31 |
2014 |
4.4 |
16.8 |
2.35 |
201.46 |
22.19 |
27.82 |
2015 |
7 |
8.05 |
2.2 |
196.94 |
22.4 |
30.47 |
2016 |
6.9 |
14.05 |
2 |
193.66 |
22.63 |
32.11 |
2017 |
5.35 |
5 |
1.7 |
190.8 |
22.88 |
33.51 |
2018 |
6.45 |
5.55 |
2.05 |
188.43 |
23.3 |
35.04 |
2019 |
6.35 |
6.95 |
1.95 |
187.82 |
23.35 |
35.4 |
2020 |
5 |
4.4 |
1.8 |
188.56 |
23.54 |
34.45 |
2021 |
5 |
13.15 |
1.95 |
189 |
23.52 |
33.52 |
Min |
4.4 |
2.65 |
1.7 |
187.82 |
22.1 |
22.58 |
Max |
7 |
16.08 |
2.65 |
211.67 |
23.54 |
35.4 |
Mean |
5.7 |
9.14 |
2.10 |
196.61 |
22.76 |
30.52 |
- Six variables (croplands, grasslands, open water, temperature, precipitation and wind velocity) were used to explain the dynamics of population sizes from 2010 to 2021. The nature of the six variables should be provided in a table, including this mean values, minimum and maximum values, units, etc.
Response: Please refer to our previous reply.
I also have some minor comments:
- Line 25. “most foraging occurred in fields (59.5%)”. What is “fields”. You may mean croplands.
Response: we modified it.
- Line 90. After listing the seven roost sites, you’d better insert the word “(Figure 1)” here, so that readers can find the sites in the figure.
Response: we modified it on page 3 line 93.
- Lines 125-126.”. I only saw three types.
Response: we modified the text on page 3 lines 131 as follow:” “three land use types were delineated in Caohai: croplands, grasslands and open water”
- Lines 132-133. “To avoid model overfitting, we used spearman's rank correlation coefficients…”. Using correlation coefficients is not a good way for checking the independence of variables, because they only represent the relationship of two variables. The best method is using variance inflation factor (VIF), which quantifies the association of a variable with all other variables. The vif() function in R package “car” can do this job.
Response: we modified the text on page 3, lines 139-142 as follow:” To avoid model overfitting, we used spearman's rank correlation coefficients the relationships among these six factors and variance inflation factor (VIF)(Supplementary S2-S3).”
|
Temperature |
Precipitation |
Wind velocity |
Cropland |
Open water |
Grassland |
VIF |
1.77 |
1.86 |
5.15 |
311.3 |
7.71 |
270.55 |
VIF<10 |
TRUE |
TRUE |
TRUE |
FALSE |
TRUE |
FALSE |
Supplementary S3 variance inflation factor (VIF) of the six factors
A VIF ≥10 indicates potentially harmful collinearity (Franke 2010). We choose factors that the VIF <10.
- Line 153. The duration (starting day and ending day) of the tracked points should be provided. The number of tracked individuals should be introduced.
Response: we modified the text on page 5 line 163 as follows:“ Figure 1. Location of 19 individuals bar-headed geese over a 13-month period in Caohai, with land use types indicated on the map”.
- Figure 3 shows the effect size of the regression coefficient of open water areas. The areas of croplands maybe removed because of its high correlation with areas of open water. The authors should clarify this point. At line 172, change “Figure 4” to “Figure 3”.
Response: we modified it.
- Figure 4. The authors should clarify these are univariate analysis using 11 years as replications.
Response: we modified the text on page 8 line 189-190 as follow:” Figure 4. the effect of Cropland, Grassland, Precipitation, Temperature, Open water, Wind velocity on 11 years wintering population size for bar-headed geese, respectively.”
Reference:
Franke, G. R. Multicollinearity. In Wiley International Encyclopedia of Marketing (eds J. Sheth and N. Malhotra).2010. https://doi.org/10.1002/9781444316568.wiem02066

Reviewer 2 Report
This is an interesting study but requires a bit of work in terms of restructuring to help the reader understand how the different components of this work fit together
some more specific comments below:
Line 10: reword this sentence it does not read well
Lines 74 – please summarise at the end of the introduction the three components of this study (direct counts, satellite tracking and feeding behaviours) and how they relate and complement each other to answer your research question/objective
Line 17: have had a positive impact
Lines 88-92- how many times were direct surveys undertaken in each year, did numbers change throughout the season?
Line 104- you say here you calculated the intensity of habitat selection, it is not clear if you present this data anywhere
Line 107 – this section on feeding behaviour could probably be a new sub-section – how long did you conduct these surveys for? And were they conducted at different times of the day and night?
Results - add subsections to describe the results of the different components of this study
Line 167 years
Results- There is no mention of the satellite tracking in the results section – it is only mentioned in the discussion
What data does Figure 1 present- is it data derived from direct counts or satellite tracking or both?
Just out of interest were there any other factors that could have also influenced the increase in population, ie, were their any changes in management, protection in the summer grounds (where are there summer grounds?), had hunting pressure (if present) been reduced? Or anything else that may also be related
Author Response
Reviewer-2
- This is an interesting study but requires a bit of work in terms of restructuring to help the reader understand how the different components of this work fit together some more specific comments below:
Response: Thank you for these positive remarks. We have made every effort to respond to your concerns.
- Line 10: reword this sentence it does not read well
Response: we modified the text on line 10 as follow:” Changes in land use changes caused by human activities can potentially result in population decline and/or local extinction.”
- Lines 74 – please summarise at the end of the introduction the three components of this study (direct counts, satellite tracking and feeding behaviours) and how they relate and complement each other to answer your research question/objective
Response: we modified the text as follow: “We used direct counts; satellite tracking and behavior observation was to explore how bar-headed geese utilize different land types and the impact of land use changes on their population dynamics.”
- Line 17: have had a positive impact
Response: we modified it.
- Lines 88-92- how many times were direct surveys undertaken in each year, did numbers change throughout the season?
Response: we modified the text on page 2 lines 88-91 as follow:” We systematically surveyed wintering populations of bar-headed geese at seven sites in Caohai every month during the wintering season for 11 consecutive years (2011-2021).” They have large population changes in November and March, and are more stable in other months.
- Line 104- you say here you calculated the intensity of habitat selection, it is not clear if you present this data anywhere
Response: we present the data in results, as follow:” The highest habitat selection intensity was recorded for bar-headed geese in grasslands (2.3), followed by croplands (1.3) and open water (1.2).”
- Line 107 – this section on feeding behaviour could probably be a new sub-section – how long did you conduct these surveys for? And were they conducted at different times of the day and night?
Response: We modified the text on page3 lines 113-114 as follow:” We used telescopes during the winter to observe behavior in daytime during the wintering season.”
- Results - add subsections to describe the results of the different components of this study
Response: We modified it.
- Line 167 years
Response: we modified it.
- Results- There is no mention of the satellite tracking in the results section – it is only mentioned in the discussion
Response: we modified the text on page 4 line 155 as follow:” We recorded a total of 45,920 bar-headed geese through satellite tracking,”
- What data does Figure 1 present- is it data derived from direct counts or satellite tracking or both? Just out of interest were there any other factors that could have also influenced the increase in population, ie, were their any changes in management, protection in the summer grounds (where are there summer grounds?), had hunting pressure (if present) been reduced? Or anything else that may also be related
Response: we modified the text on page 5 line 161-162 as follow:” Figure 1. Location of 19 individuals bar-headed geese derived from satellite tracking over a 13-month period in Caohai, with land use types indicated on the map.
On page 9 lines 231-235 as follow:” In addition, the bar-headed geese wintering in Caohai mainly breed in Zoige. Between 2000 and 2019, the Zoige wetlands were generally improved via several eco-logical restoration projects and protection [management (Yang et al., 2021). Caohai National Nature Reserve has strengthened wetland management since the promulgates of the ‘Regulations on Wetland Protection in Yunnan Province’ in 2013. These factors will also play a positive role in increasing the overwintering population of bar-headed geese.
Reference
Yang, J.; Yuan, Y.; Wang, B.; Wang, S. Remote Sensing Analysis of Ecological Environment of ruoergai Wetland Ecological Protection red Line Area. Sichuan Environment. 2021, 40, 170-184.(in Chinese)
